# Effect of Liquid Marble 3D Culture System on In Vitro Maturation and Embryo Development of Prepubertal Goat Oocytes

**DOI:** 10.3390/ani15020188

**Published:** 2025-01-12

**Authors:** Andrea Podda, Linda Dujíčková, Federica Ariu, Giovanni Giuseppe Leoni, Dolors Izquierdo, Maria-Teresa Paramio, Luisa Bogliolo

**Affiliations:** 1Department of Veterinary Medicine, University of Sassari, 07100 Sassari, SS, Italy; a.podda1@studenti.uniss.it (A.P.); ldujickova@uniss.it (L.D.); federica@uniss.it (F.A.); 2Department of Biomedical Science, University of Sassari, 07100 Sassari, SS, Italy; gioleoni@uniss.it; 3Department of Animal and Food Science, Faculty of Veterinary Medicine, Universitat Autònoma de Barcelona (UAB), 08193 Barcelona, Spain; dolors.izquierdo@uab.cat (D.I.); teresa.paramio@uab.cat (M.-T.P.)

**Keywords:** 3D culture system, liquid marble, prepubertal goat, oocytes, embryo development

## Abstract

The three-dimensional in vitro maturation of oocytes is a promising approach that can be used to improve in vitro embryo production efficiency. There are currently few studies on this and there are no standardized three-dimensional in vitro maturation systems. In this study, we evaluated the potential beneficial effects of liquid marble microbioreactors as three-dimensional culture systems for the in vitro maturation of prepubertal goat oocytes. We found that the liquid marble system did not affect meiotic progression, the density of transzonal projections, nor the reactive oxygen species and glutathione levels of oocytes in comparison to two-dimensional in vitro maturation systems. However, in vitro maturation in the liquid marble system increased mitochondrial activity and modified the distribution of these organelles of matured oocytes. After the parthenogenetic activation of oocytes, embryo development up to the blastocyst stage was not affected by the liquid marble system. The liquid marble microbioreactor should be considered as a potential alternative to two-dimensional systems for the in vitro maturation of prepubertal goat oocytes.

## 1. Introduction

Oocyte in vitro maturation (IVM) is a major step during in vitro embryo production (IVEP) as it involves both the progression of the meiotic cycle and cytoplasmic reorganization, which are essential for the oocyte’s ability to undergo fertilization and to support early embryonic development [1,2]. In ruminants, even if high rates of maturation of cumulus oocyte complexes (COCs) can be obtained, their developmental competence is still suboptimal [3], as evidenced by their low development up to the blastocyst stage and poor viability after being transferred into recipient animals. Oocytes retrieved from prepubertal donors can be used for IVEP to reduce the generation interval and thereby increase the rate of genetic gain [4,5].

However, IVEP efficiency using oocytes from prepubertal donors is reduced compared to that in adult donors [6,7,8]. IVEP success is mainly determined by the intrinsic quality of the oocytes and the IVM culture systems. Over the last few decades, several attempts have been made in different species to identify the appropriate culture environment for the IVM of oocytes, including the formulation of new media and supplementation with various compounds, such as follicular fluid, antioxidants, cytokines, growth factors and hormones [9,10,11,12,13]. More recently, bioengineering approaches designed to mimic physiological follicular structures have attracted substantial interest [14]. These strategies aimed to maintain the three-dimensional (3D) structure of COCs to facilitate bidirectional communication between the oocytes and the surrounding granulosa cells.

A variety of matrices and culture techniques, such as agarose matrix [15], glass scaffolds [16] and alginate microbeads [17], have been developed for the 3D IVM culture of oocytes in different species. These techniques avoid the disadvantages of 2D systems, such as COC flattening at the bottom of the culture, which greatly decreases the amount of the cell surface exposed to the media [18].

Liquid marbles (LMs) are a form of 3D microbioreactor first described by [19], consisting of a small drop of liquid encapsulated by hydrophobic powder particles. The particles adhere to the liquid droplet, creating a closed microenvironment, while allowing gas exchange between the liquid inside and the surrounding environment. LMs have been widely used in cell cultures to support the growth of embryonic stem cells [20,21], fibroblasts [22] and red blood cells [23]. According to these studies, LMs may be a viable technique for oocyte IVM since they facilitate gas exchange, minimize exposure to harmful chemicals and create a stable microenvironment. Thus far, it has been evidenced that the LM microbioreactor is a practical method that offers an appropriate microenvironment for the IVM of sheep [24] and pig oocytes [25]. In prepubertal ovines, there was an increase in the rate of blastocysts from oocytes matured in the LM microbioreactor compared to conventional maturation, despite similar nuclear maturation rates [26]. On the contrary, the IVM of bovine oocytes in LMs led to reduced embryo development [27].

According to our knowledge, there is no study using LM microbioreactors for the IVM of prepubertal goat oocytes. Therefore, the aim of this study was to evaluate the potential effect of LM microbioreactors as a 3D culture system to mature in vitro prepubertal goat oocytes in comparison to a two-dimensional (2D) culture system in drops under mineral oil. For such a purpose, the feasibility of the LM system was tested by evaluating the in vitro meiotic competence of oocytes and embryo development up to the blastocyst stage, following parthenogenetic activation as a commonly used method in goat oocyte research [28,29]. Moreover, we examined oocyte cumulus cell–oocyte communication (the density of transzonal projections), oxidative homeostasis (ROS and GSH levels) and the mitochondrial state, which are important functional markers linked to the quality of oocyte maturation.

## 2. Materials and Methods

All chemicals were purchased from Sigma-Aldrich Chemical Co., Ltd. (St. Louis, MO, USA), unless otherwise specified.

### 2.1. Oocyte Collection and In Vitro Maturation

Ovaries from prepubertal goats (1 to 2 months old) were collected at a local slaughterhouse and transported to the laboratory at 35–37 °C in phosphate-buffered saline (PBS). COCs were recovered by slicing the ovaries in HEPES-buffered (25 mM) TCM-199 supplemented with 2.2 mg/mL NaHCO_3_, 50 mg/mL gentamycin and 11.1 mg/mL heparin. COCs with several layers of compact cumulus cells (CCs) and homogeneous cytoplasm were selected for IVM and randomly divided between two different IVM systems: the control group (CTR) COCs were cultured in 30 µL drops of IVM medium covered with mineral oil (10 COCs/drop, Figure 1A); the liquid marble group (LM; Figure 1B) COCs were in vitro maturated in groups of 10 COCs in drops of 30 μL of maturation medium, coated by a hydrophobic fumed silica powder (CAB-O-SIL^®^TS-5309, Cabot Corporation, Tuscola, IL, USA). The preparation of the LM microbioreactor is described in Figure 1B–G, according to a previous study [24]. Briefly, a drop of maturation medium with COCs was aspirated by pipette and placed in a 60 mm Petri dish with hydrophobic powder. It is necessary to gently roll the drop several times in the powder to fully cover it. Afterwards, the drop of medium was aspirated by a Pasteur pipette and placed in a 35 mm Petri dish. Then, three 35 mm Petri dishes with drops were placed in a 90 mm Petri dish containing sterile water (8–10 mL) to increase humidity and avoid dehydration.

COCs of both groups were cultured for 24 h at 38.5 °C in humidified air with 5% CO_2_ in TCM-199 supplemented with 0.2 mM sodium pyruvate, 3.7 µM E2, 10 ng/mL epidermal growth factor (EGF), 1 mM glutamine, 5 µg/mL gentamicin 10% of fetal bovine serum (FBS), 5 μg/mL LH and 5 μg/mL FSH.

### 2.2. Evaluation of Oocyte Nuclear Stage

After IVM, denuded oocytes were fixed in ethanol: acetic acid (3:1) overnight at 4 °C and then mounted on a slide in 3 µL of glycerol solution containing Hoechst 33258 (1 µg/mL; Invitrogen, Eugene, OR, USA). They were then covered with a wax-supported coverslip. Finally, the nuclear stage of oocytes was analyzed by fluorescence microscope (Olympus BX50) and classified as follows: germinal vesicle (GV), germinal vesicle breakdown (GVBD), metaphase I (MI), metaphase (MII) or degenerated.

### 2.3. Assessment of Transzonal Projections

Transzonal projections (TZPs) were stained by phalloidin-FITC (Invitrogen) staining at 0 h (T0), 6 h (T6) and 24 h (T24) of IVM. COCs were partially denuded from CCs (2–4 layers of cumulus cells) and fixed in 4% paraformaldehyde (PF; *w*/*v*) for 20 min at room temperature (RT), then permeabilized in 0.25% Triton X-100 for 30 min and finally incubated in 5 µg/mL phalloidin-FITC for 60 min. After this procedure, COCs were mounted on glass slides with 10 μg/mL Hoechst 33258 in PBS and glycerol solution. Then, images were taken under a laser-scanning confocal microscope (LSCM, Leica TCS SP5, Wetzlar, Germany). For the images, we used a 40× magnification under mineral oil (phalloidin-FITC: λ_ex_ 488 nm, λ_em_ 520 nm; Hoechst: λ_ex_ 405 nm and λ_em_ 465 nm). TZPs were observed as continuous filaments going from CCs to the oocyte. The TZPs density was quantified using ImageJ software (Version 1.51 h; National Institute of Health, Bethesda, MD, USA) by measuring the intensity of the relative fluorescence in the zona pellucida area delimited by the polygon selection tool and normalized to that of control oocytes. The intensities of signals were expressed as arbitrary units (AU).

### 2.4. Measurement of Reactive Oxygen Species (ROS) and Glutathione (GSH) Levels

Intra-oocyte ROS and GSH levels were measured by staining with 2′,7′-dichlorodihydrofluorescein diacetate (H_2_DCF-DA; Molecular Probes Inc., Eugene, OR, USA) and CellTracker Blue (CMF2HC, Molecular Probes Inc.). Briefly, oocytes were denuded and incubated for 30 min with either 10 μM H_2_DCF-DA or 10 μM CellTracker Blue in 0.3% BSA-PBS at 38.5 °C. After washing them in 0.1% BSA-PBS, each of the oocyte was transferred to a slide in a 50 μL droplet of PBS-PVA and observed using an epifluorescence microscope (Olympus IX 70, Tokyo, Japan) with UV filters (460 nm for ROS and 370 nm for GSH). The average fluorescence intensity per oocyte was measured using Image J software (Version 1.51 h) and normalized to that of control oocytes. The intensities of signals were expressed as arbitrary units (AU).

### 2.5. Evaluation of Mitochondrial Activity and Distribution

At 0 h (T0) and 24 h (T24) of oocyte IVM, the oocytes were denuded and stained with MitoTracker Orange CMTM Ros (Molecular Probes, Inc.) to detect mitochondrial activity and distribution. Briefly, oocytes were incubated for 30 min at 38.5 °C in humidified air with 5% CO_2_ in PBS-BSA containing 200 nM MitoTracker Orange CMTM Ros. Next, oocytes were mounted on glass slides with 10 μg/mL Hoechst 33258 in PBS and glycerol solution, covered by a coverslip and, finally, imaged under a LSCM. A helium/neon laser ray at 543 nm, equipped with 551 nm (excitation) and 576 nm (emission) filters, was used to detect the MitoTracker Orange CMTM Ros. LSCM settings were kept constant for all experiments. In each individual oocyte, the MitoTracker fluorescence intensity was measured on the section corresponding to the oocyte equatorial plane using ImageJ software. The intensities of signals were normalized to those of control oocytes and were expressed as AU.

Mitochondrial distribution was evaluated according to the study [30] based on occurring patterns: diffused (mitochondria distributed homogeneously through the cytoplasm), semi-peripheral (mitochondria distribution between peripheral and diffused) or peripheral (mitochondria distributed mostly in the cortical region of the oocyte).

### 2.6. Parthenogenetic Activation of Oocytes and Embryo Culture

After IVM, oocytes were activated with 5 μM ionomycin in PBS for 5 min at 38.5 °C followed by 4 h culture in TCM-199 with 2 mM 6-dimethylaminopurine (6-DMAP) in a humidified atmosphere with 5% CO_2_ at 38.5 °C. After that, oocytes were cultured for 8 days in 10 µL drops of BO-IVC medium (IVF Bioscience; Cornwall, UK) under mineral oil (SAGE^TM^, Cooper Surgical, Trumbull, CT, USA, lot no. ART-4008-5P) in a humidified atmosphere with 5% CO_2_, 5% O_2_ and 90% N_2_ at 38.5 °C. The cleavage was evaluated morphologically after 24 h and 30 h of culture, while development to the blastocyst stage was evaluated on days 7 and 8.

### 2.7. Assessment of Blastocyst Cell Number

Blastocysts were incubated in TCM-199 with 1% Triton X-100 (*v*/*v*) and 100 µg/mL propidium iodide (PI) for 25 s. Afterwards, blastocysts were transferred to pure ethanol with 25 µg/mL Hoechst 33258 and stored at 4 °C overnight. Blastocysts were then transferred onto a glass slide in a drop of Vectashield^®^ Mounting Medium (©Vector Laboratories, Burlingame, CA, USA), covered with a coverslip and finally evaluated by epifluorescence microscope (Olympus BX50; λ_ex_ 370 nm and λ_em_ 465 nm). A digital image of each embryo was taken, and the numbers of inner cell mass (ICM, blue) and trophectoderm (TE, red) nuclei were counted by ImageJ software (Version 1.51 h).

### 2.8. Statistical Analysis

All statistical analyses were performed using STATA\IC 11.28 (StataCorp LLC, College Station, TX, USA). Data on nuclear maturation, mitochondrial distribution, cleavage, blastocyst formation and blastocyst morphological classification were analyzed by chi-square tests. Blastocyst cell number, mitochondria activity, ROS levels, GSH levels and TZPs density were analyzed with two-way ANOVA after the Levene‘s test to assess equal variances followed by Tukey’s multiple-comparison tests. Differences with *p* < 0.05 were taken into consideration as statistically significant. All experiments were replicated at least three times.

## 3. Results

### 3.1. Effect of LM System on Nuclear Maturation

The results of IVM are summarized in Figure 2. A total of 138 oocytes (*n* = 69, CTR; *n* = 69, LM) from prepubertal goat ovaries were used for IVM. In the CTR group, 52,17% of oocytes reached the MII stage, while 20.29% stayed in GV and 17.39% were in the GVBD/MI stage. The rest of the oocytes (10.14%) were degenerated. In the case of the LM group, 44.12% of oocytes were in the MII stage, 7.35% in GV and 29.41% in the GVBD/MI stage. Degenerated oocytes after 24 h of IVM represented 19.12%. According to these results, only the percentage of GV oocytes was significantly increased in the CTR group compared to the LM group.

### 3.2. Effect of LM System on Transzonal Projections Density

Based on the relative fluorescence intensity of TZPs, the communication between CCs and oocytes was determined (Figure 3A, B). During IVM, we observed a significant decrease in relative fluorescence intensity in TZPs after 24 h of IVM in both the CTR (T24 CTR, 0.55 ± 0.08 AU; *n* = 13) and the LM group (T24 LM, 0.37 ± 0.04 AU; *n* = 12), compared to the CTR immature group (T0, 1 ± 0.12 AU; *n* = 12). Significant decreases were also observed between 6 h (T6 CTR, 1.17 ± 0.15 AU; *n* = 12) and 24 h of maturation in the CTR group as well as the LM group (T6 LM, 0.86 ± 0.16 AU; *n* = 11). No significant differences were detected in the TZPs density between the CTR and LM groups at T6 and T24 IVM.

### 3.3. Effect of LM System on ROS and GSH Levels

Based on the relative fluorescence intensity (Figure 4), no significant differences were detected between ROS levels in the CTR (1 ± 0.05 AU; *n* = 22) and LM (0.85 ± 0.06 AU; *n* = 13) groups. The LM culture system did not affect the GSH levels (1.02 ± 0.06 AU; *n* = 13) compared to the CTR group (1 ± 0.05 AU; *n* = 22).

### 3.4. Effect of LM System on Mitochondrial Status

Based on the relative fluorescence intensity, mitochondrial activity (Figure 5) was significantly increased (*p* < 0.01) in the T24 LM group (1.14 ± 0.08 AU; *n* = 21) compared to T24 CTR (0.55 ± 0.10 AU; *n* = 20), and it did not show statistical differences with the T0 group (1 ± 0.09 AU; *n* = 21). Mitochondrial activity in the T24 CTR group significantly (*p* < 0.01) decreased after IVM compared to the T0 group.

The distribution of mitochondria was classified in three patterns: diffused, semi-peripheral and peripheral (Figure 6). The distribution patterns (Table 1) were statistically different among groups (*p* < 0.05). The diffused distribution of mitochondria was lower in T0 and T24 CTR compared to T24 LM (47.62%, 45.00% and 90.47%, respectively, in T0, T24 CTR and T24 LM). The semi-peripheral distribution represented 38.10% in T0 and 40.00% in the T24 CTR group and 9.52% in the T24 LM group. The peripheral distribution was comparable in the T0 (14.28%) and T24 CTR (15.00%) groups, but it did not occur in the T24 LM group.

### 3.5. Effect of LM System on Embryo Development and Blastocyst Cell Number After Parthenogenetic Activation

Cleavage and blastocyst rates did not statistically differ between LM and CTR groups (Figure 7A). From a total of 188 parthenogenetically activated oocytes in the CTR group, 42.02% cleaved after 24 h and 63.30% cleaved after 30 h. The blastocyst rate after 8 days was 10.64%. In the LM group (*n* = 198), the cleavage rate was 39.90% at 24 h and 58.59% at 30 h. The blastocyst rate after 8 days was 10.10%.

The distribution of blastocysts according to their developmental stage (Figure 7B) did not show significant differences between the CTR and LM groups. In the CTR group, 35% were classified as normal blastocysts, 25% as expanded, 10% as hatching and 30% as hatched. In the LM group, the distribution of blastocysts was 20% normal, 30% expanded, 10% hatching and 40% hatched.

The mean total cell number (TCN), inner cell mass and trophectoderm cells showed no differences between blastocysts obtained in the CTR and LM groups (Table 2).

## 4. Discussion

For the first time, in this study LM was tested in prepubertal goat IVM oocytes. We showed that there were no differences between 2D and 3D systems regarding the oocyte maturation rate, nor in the development up to the blastocyst stage or in the embryo quality, nor ROS and GSH in oocytes. However, we found increased mitochondrial activity and a different organization of these organelles after maturing oocytes in an LM system.

Initially, we evaluated the effect of LM on nuclear maturation. Despite the fact that the proportion of GV oocytes was significantly higher in the CTR (20.29%) compared to the LM group (7.35%), the proportion of MII oocytes after IVM did not change in LM (44.12%) when compared to the CTR group (52.17%). The first study [24] using LM in sheep oocytes did not obtain differences in maturation and blastocyst rates. Later, these authors obtained a significant increase in blastocyst rate using the LM system (28.26% vs. 14.06% in CTR) but no effect on the maturation rate of prepubertal ovine oocytes [26]. Similarly, in bovines [27], other authors did not observe an effect of LM on the maturation rate. However, they described a lower expansion of CCs and even lower blastocyst rates together with total cell numbers. In our study, the expansion of CCs remained comparable in both groups, as well as the blastocyst rate (10.10% in LM vs. 10.64% in CTR) and blastocyst cell number.

Next, we investigated ROS and GSH levels in oocytes cultured in 2D and 3D systems and observed that the LM system did not increase oxidative stress or alter the GSH level. The effect of the LM system on ROS and GSH levels has never been demonstrated. While low and balanced production of ROS is normal for adequate oocyte survival and subsequent embryo development [31], increased ROS production can induce oxidative stress, leading to cell damage. To avoid this, cells have antioxidant networks to scavenge excessively produced ROS and create homeostasis [32]. Glutathione, as an exogenous antioxidant, not only maintains the intracellular redox balance but also plays an important role in fertilization and early embryo development [33]. In oocytes, supplying GSH is dependent upon cumulus cells through cell-to-cell communication via transzonal projections [34].

We did not observe significant differences in the TZPs relative fluorescence intensity between the two systems of IVM. Transzonal projections are crucial structural elements of the follicle as they provide the site of gap junctional communication and allow other interactions between oocyte and cumulus cells [35]. In our study, the fluorescence intensity of F-actin in TZPs decreased gradually during IVM in both 2D and 3D groups. A decrease in TZPs fluorescence intensity during IVM was also observed in porcine oocytes [36,37]. The loss of the TZPs during maturation appears to be important for optimal oocyte quality [38]. In human, the authors of [39] observed that some transzonal projections were still present in COCs after IVM, which is in line with our results, as relative fluorescence intensity was significantly decreased compared to immature oocytes.

An interesting finding of our study concerns the effect of the LM system on the mitochondrial status in matured oocytes. Indeed, our results demonstrate that IVM in LM system resulted in an increase in the mitochondrial intensity. Moreover, the 3D system enhanced the rate of homogeneous distribution of mitochondria in the oocyte cytoplasm. To the best of our knowledge, this is the first study to assess how mitochondria behave in oocytes that have matured in the LM system. In another study [40], increased mitochondrial activity and membrane potential were observed in porcine oocytes using alginate-hydrogel embedding as a 3D culture system. However, they did not observe changes in the mitochondrial distribution.

Mitochondria have the essential role of producing ATP via oxidative phosphorylation, which is required for the oocyte to mature and for the fertilized oocyte to proceed through development [41]. In addition to producing most of the oocyte’s ATP, these organelles play a key role in controlling Ca^2+^ and redox homeostasis [42]. Mitochondria are the most studied organelles relative to developmental competence and their activity and distribution are considered markers for the evaluation of oocyte cytoplasmic maturity [43]. Changes in mitochondrial distribution, from a peripheral location to a more uniform distribution throughout the oocyte cytoplasm, have been observed during maturation in different animal species [43,44]. A dispersed homogeneous distribution of mitochondria is assumed to be a sign of a matured cytoplasm, while a peripheral distribution of mitochondria characterizes meiotically incompetent oocytes [45]. In prepubertal goats, mitochondria are distributed in the peripheral zone, polarized to the region opposite to the metaphase spindle within the polar body, in 86.5% of mature oocytes [46]. In contrast, in oocytes from adult goats, this distribution occurs in 50% of MII oocytes. Prepubertal oocytes have been classified as oocytes with a lower developmental competence than adult ones [46,47] and the lower rate of peripheral clustered mitochondria in adult ones suggests that this distribution is related to the lower competence of the oocyte. Mitochondrial distribution is also related to mitochondrial activity and the amount of ATP in ovine oocytes [47]. Our results demonstrate that the LM system is able to induce changes in mitochondrial distribution (from a semi-peripheral/peripheral pattern to a diffused pattern) during IVM while also increasing mitochondrial activity. It is possible that both increased activity and the different distribution of mitochondria in LM group could account for a decreased proportion of the oocytes that remain in the GV stage. However, according to the mitochondrial status, our findings suggest that IVM with LM may enhance the cytoplasmic maturity of prepubertal goat oocytes. These changes in mitochondria activity were not reflected in a positive effect in oocyte nuclear maturation or embryonic development.

Further studies are needed to better understand how the LM system affects embryo quality and to correlate changes in mitochondrial status with possible effects on embryonic development. Analyzing the expression of particular genes linked to the quality of embryos based on the culture environment (control or LM) may be one way to address this issue.

The use of 3D LM microbioreactors as a culture system for oocyte IVM has several advantages. It provides a stable microenvironment for IVM, enabling the culture of COCs without the use of mineral oil and in a minimal volume of medium. In fact, the use of mineral oil during IVM has been associated with delayed nuclear maturation in porcine oocytes [48] and meiosis progression in mouse oocytes [49]. Mineral oil can have a negative effect on embryonic developmental competence, possibly releasing embryotoxic molecules into the culture medium like peroxides, alkenals, aldehydes, Triton X-100 and zinc [50] which, in turn, affect the embryo’s development [51,52,53]. In addition, by using the LM system, the possibility of contact of the media with the plastic can be reduced. While the usage of plastics is common in laboratories, evidence demonstrates that this may have negative effects on the gametes [54]. Therefore, the implementation of the LM system could be an effective method to avoid contact with plastics. Another advantage of the LM system is that unlike other 3D culture systems, the generation of LM beads does not require solid supports such as hydrogels, scaffolds or matrices and culture equipment to obtain a 3D arrangement.

It is important to coat LM drops with enough powder by rolling them into the powder multiple times to prevent them from breaking during IVM. Furthermore, the LM size of the drops should not exceed 30 µL of IVM media, and sterile water (8 to 10 mL) should be added to a 90 mm Petri plate, to guarantee enough humidity throughout the culture process, in order to prevent evaporation.

The implementation of the LM system has potential for use in other steps of the IVEP procedure, such as oocyte fertilization and individual embryo cultures. Indeed, some preliminary studies of our research group have found that using LM for a biphasic IVM of lamb oocytes can prolong the persistence of CC–oocyte communication and reduce oxidative stress in oocytes (unpublished data). To conclude, our findings have demonstrated that 3D LM cultures are a simple, fast and low-cost method that provides a suitable 3D microenvironment to induce prepubertal goat oocyte IVM. Nevertheless, further studies are required, and optimization of the 3D culture system is necessary in order to draw a conclusive evaluation of its potential not only during IVM, but also in fertilization and embryo cultures.

## 5. Conclusions

For the first time, a LM culture system has been tested in goat IVM. The results showed that there is no difference in the nuclear maturation and development of embryos resulting from oocytes cultured in the 3D system compared with the 2D system. No adverse effect was observed in terms of alterations in CC–oocyte communication, ROS production or GSH levels, while a potential positive effect on the mitochondria is to be further studied and better explored. LM microbioreactors have proven to be a suitable alternative to the traditional 2D culture system for the IVM of prepubertal goat oocytes. Additional research is still needed to validate this culture method for broader use in IVEP procedures.

## Figures and Tables

**Figure 1 animals-15-00188-f001:**
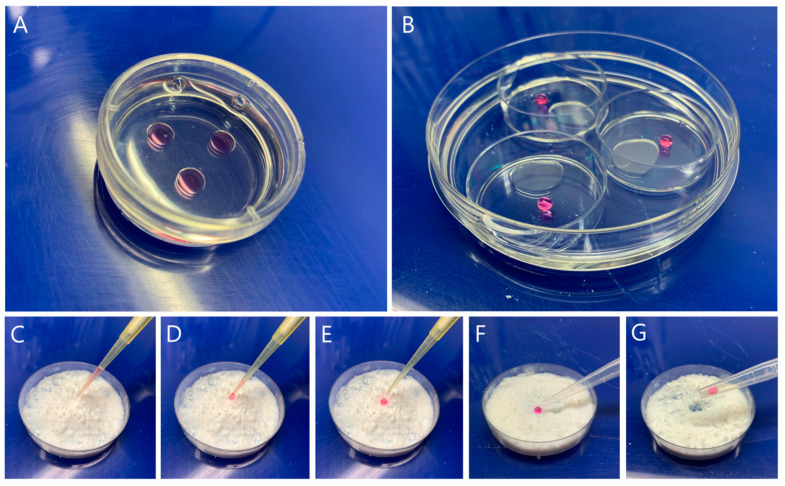
Schematic picture of IVM culture systems. (**A**) 2D system with three drops of maturation medium containing oocytes covered with mineral oil. (**B**) The LM system—three 35 mm Petri dishes with LM droplets containing oocytes, placed in a 90 mm Petri dish with sterile water, to ensure enough humidity during the culture. (**C**–**G**) Preparation of LM droplets. (**C**) A drop of maturation media containing oocytes is being aspirated with a pipette. (**D**) The drop is slowly released into the dish with the powder. (**E**) After release, gently rolling of the drop in the powder is required to fully cover the drop. (**F**) A well-coated drop of maturation medium with the hydrophobic powder in the Petri dish. (**G**) The drop is aspirated by a Pasteur pipette and placed into the 35 mm Petri dish.

**Figure 2 animals-15-00188-f002:**
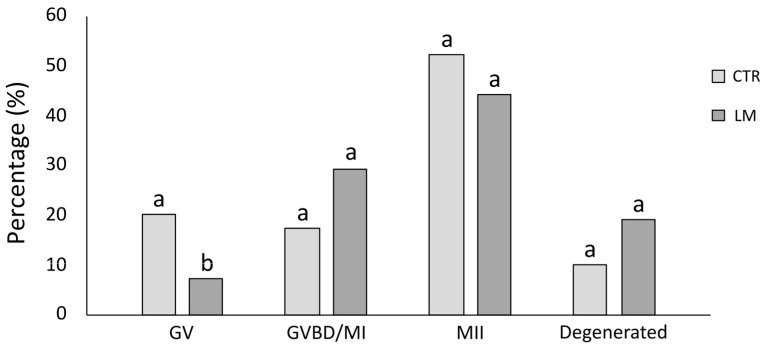
Effect of the LM system on the meiotic maturation of prepubertal goat oocytes. GV: germinal vesicle, GVBD: germinal vesicle breakdown, MI: metaphase I, MII: metaphase II. Different superscripts indicate significant differences at *p* < 0.05.

**Figure 3 animals-15-00188-f003:**
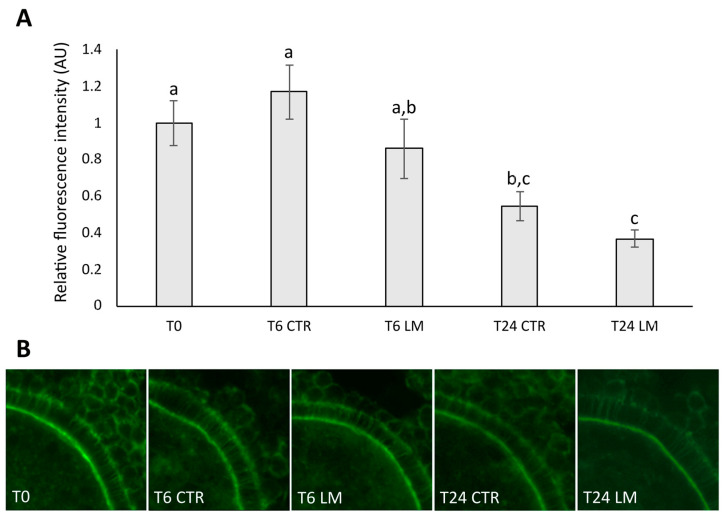
Effect of the LM system on the TZPs density of prepubertal goat oocytes. (**A**) Phalloidin-FITC average fluorescence intensity in the zona area of COCs after recovery (T0), 6 h (T6) and 24 h (T24) of IVM. (**B**) Representative confocal images of oocytes with stained TZPs (actin, green) which appear as continuous filaments going from the CCs to the oocyte through the zona region. Values are expressed as the mean ± SEM. Different superscripts indicate significant differences at *p* < 0.05.

**Figure 4 animals-15-00188-f004:**
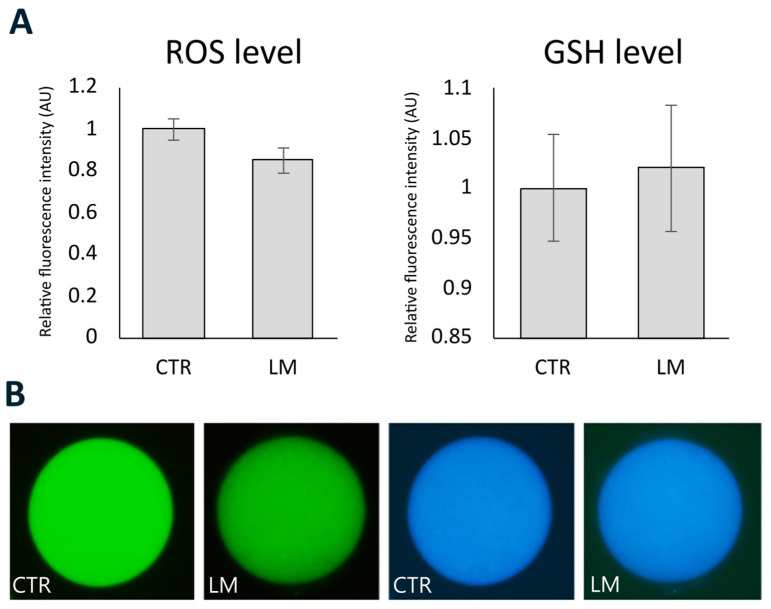
Effect of the LM system on (**A**) the intracellular ROS and GSH levels of prepubertal goat oocytes based on the relative fluorescence intensity. (**B**) Epifluorescence photomicrographs of oocytes stained with H_2_DCF-DA and CellTracker Blue to detect ROS (green) and GSH (blue) levels respectively, in the tested groups. Values are expressed as the mean ± SEM.

**Figure 5 animals-15-00188-f005:**
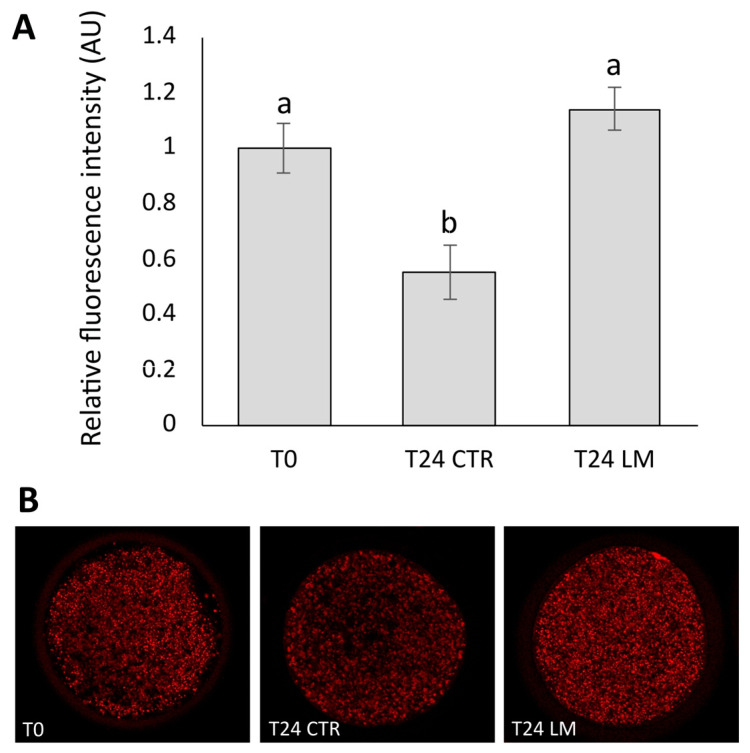
(**A**) Effect of the LM system on mitochondrial activity in prepubertal goat oocytes. (**B**) Representative images of mitochondrial activity in oocytes. Values are expressed as the mean ± SEM. Different superscripts indicate significant differences at *p* < 0.01.

**Figure 6 animals-15-00188-f006:**
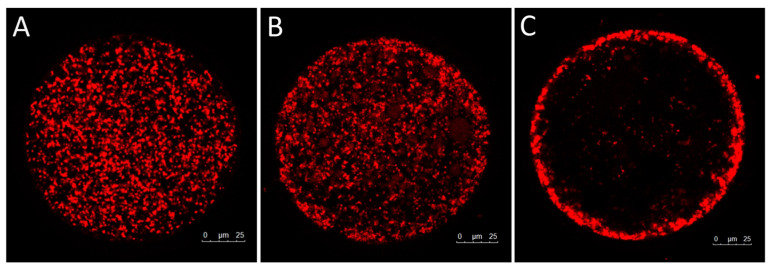
Representative images of distribution patterns of mitochondria (red signal) in goat oocytes: (**A**) diffused, (**B**) semi-peripheral and (**C**) peripheral.

**Figure 7 animals-15-00188-f007:**
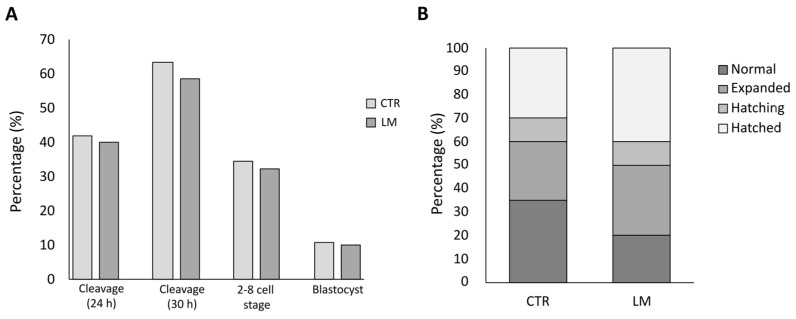
Effect of the LM system on embryo development. (**A**) Cleavage at 24 and 30 h post activation, and the blastocyst rate after 8 days from activation. (**B**) Proportion of normal, expanded, hatching and hatched blastocysts on day 8 of culture in the tested groups. Differences between groups were not significant.

**Table 1 animals-15-00188-t001:** Distribution of mitochondria in goat oocytes after IVM.

Groups	Number of Oocytes	Diffused, *n* (%)	Semi-Peripheral, *n* (%)	Peripheral, *n* (%)
T0	21	10 (47.62) ^a^	8 (38.10) ^a^	3 (14.28) ^a^
T24 CTR	20	9 (45.00) ^a^	8 (40.00) ^a^	3 (15.00) ^a^
T24 LM	21	19 (90.47) ^b^	2 (9.52) ^b^	0 (0.00) ^b^

Different superscripts in columns indicate significant differences at *p* < 0.05.

**Table 2 animals-15-00188-t002:** Total cell numbers of resulting blastocysts.

Groups	TCN, *n* ± SEM	ICM, *n* ± SEM	TE, *n* ± SEM
CTR	95.56 ± 11.03	15.81 ± 4.23	79.75 ± 10.33
LM	96.53 ± 14.38	13.20 ± 2.97	83.33 ± 13.80

TCN—total cell number; ICM—inner cell mass; TE—trophectoderm. Differences between groups were not significant.

## Data Availability

The raw data supporting the conclusions of this article will be made available by the authors on request.

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
