# Peer review of "Effect of Liquid Marble 3D Culture System on In Vitro Maturation and Embryo Development of Prepubertal Goat Oocytes"

_animals, 2025, doi:10.3390/ani15020188_

Round 1
Reviewer 1 Report
Comments and Suggestions for Authors
Review animals-3362766
Effect of liquid marble 3D culture system on in vitro maturation and embryo development of prepubertal goat oocytes
The study evaluated the effects of a three-dimensional (3D) oocyte culture system based on the liquid marble (LM) microbioreactor compared to a traditional two-dimensional (2D) system, using oocytes from prepubertal goats. The results showed that the LM system did not affect meiotic progression, levels of reactive oxygen species (ROS), glutathione (GSH), or transzonal projection density. However, the LM system significantly increased mitochondrial activity and altered mitochondrial distribution, indicating a positive impact on the cytoplasmic environment of the oocytes. Despite these changes in mitochondrial parameters, no differences were observed in embryonic developmental competence, including blastocyst formation rates and total embryo cell numbers. The approach of using three-dimensional (3D) culture systems in prepubertal oocytes is promising and aligns with current trends in bioengineering applied to assisted reproduction. The experimental protocol is clearly described, including precise details about the media used, culture conditions, and analysis methods adopted. This thoroughness is complemented by a solid theoretical foundation, referencing relevant previous studies and highlighting innovations associated with the application of 3D systems. The presentation of data is also noteworthy, being well-organized and accessible, with effective use of tables and figures to facilitate the interpretation of results and emphasize the significance of the findings.
General comments
- Figure 1. Schematic picture of IVM culture systems
- I suggest creating a schematic representation of the LM system, as the photos are not clearly conveying the operation of the system (Page 3).
- Lack of contextualization of the results:
- Although differences in mitochondrial activity and distribution were identified, the functional impact of these changes was not clearly discussed, leaving gaps regarding the biological relevance of these findings.
- Insufficient analysis of technical limitations:
- The issue of the skill required to prepare the drops coated with hydrophobic powder was briefly mentioned, but without exploring possible solutions or technical improvements (Line 382 – 384).
4. Limited expansion of the discussion:
- The discussion could be strengthened by relating the findings to known mechanisms and better exploring the practical potential impacts of the LM system in in vitro embryo production programs.
- Weak conclusions:
- The conclusions state that the LM system is an alternative to the 2D system, but the presented data (such as blastocyst rates and cell numbers) do not show significant benefits. It is necessary to revise this conclusion or better justify the potential benefits of the LM system.
- Technical language and editorial review:
- There are minor language issues, such as grammatical errors and inconsistent formatting. For example, the repetition of introductory phrases throughout the text makes the reading redundant.
- Lack of molecular biomarker analysis:
- The study could have evaluated changes in gene expression profiles in blastocysts, which would help correlate alterations in mitochondrial status with possible impacts on embryonic development.
8. Insufficient analysis of embryonic quality indicators:
- The study analyzed basic parameters (such as total blastocyst cell numbers), but lacked more in-depth indicators, such as the distribution of cells between the inner cell mass and trophoblast, or the expression of specific genes related to embryonic quality.
Final considerations
The use of three-dimensional systems in assisted reproduction is a promising and innovative field, and the proposal to use the LM system is worth attention. However, for the study to have a greater impact, it is essential that it be further developed. If the authors expand the discussion, include molecular analyses or complementary data, such as gene expression in blastocysts, and provide stronger justifications for their conclusions, the article could become more relevant. The study presents an interesting technological approach (the LM system) that, with the appropriate adjustments, has the potential to be useful in future applications.
Reviewer 2 Report
Comments and Suggestions for Authors
The study assessed the use of an alternative culture system to in vitro mature pre-pubertal goat oocyte for the first time. The LM 3-D microbioreactor is a new approach for in vitro culturing cells and embryos, and as such, research needs to provide evidence that it is a safe system and suitable to support oocyte and embryo activities. The experimental design was adequate to test the new system and the results obtained support the conclusions of the study. However, there are details, described in a separate file, that authors need to address to enhance the study contribution to the general knowledge on oocyte in vitro maturation for IVEP.

Reviewer 3 Report
Comments and Suggestions for Authors
As per the attached report

As per the attached report
Round 2
Reviewer 1 Report
Comments and Suggestions for Authors
Dear Editor,
After a detailed review of the revised manuscript, I found that the authors have made significant improvements in response to the previous comments. The revisions effectively address the critical points raised, enhancing both the clarity and the scientific rigor of the work.
These changes have strengthened the methodological robustness and the potential impact of the study in the field, making the manuscript suitable for publication. Therefore, in my opinion, the article should be accepted.
I remain available for any further clarifications.
Best regard,